# Transient Enlargement in Meningiomas Treated with Stereotactic Radiotherapy

**DOI:** 10.3390/cancers14061547

**Published:** 2022-03-17

**Authors:** Ziad Maksoud, Manuel Alexander Schmidt, Yixing Huang, Sandra Rutzner, Sina Mansoorian, Thomas Weissmann, Christoph Bert, Luitpold Distel, Sabine Semrau, Sebastian Lettmaier, Ilker Eyüpoglu, Rainer Fietkau, Florian Putz

**Affiliations:** 1Department of Radiation Oncology, Universitätsklinikum Erlangen, Friedrich-Alexander-Universität Erlangen-Nürnberg, Universitaetsstraße 27, 91054 Erlangen, Germany; ziad.maksoud@ukmuenster.de (Z.M.); yixing.huang@uk-erlangen.de (Y.H.); kn0t3n@gmx.net (S.R.); sina.mansoorian@uk-erlangen.de (S.M.); thomas.weissmann@uk-erlangen.de (T.W.); christoph.bert@uk-erlangen.de (C.B.); luitpold.distel@uk-erlangen.de (L.D.); sabine.semrau@uk-erlangen.de (S.S.); sebastian.lettmaier@uk-erlangen.de (S.L.); rainer.fietkau@uk-erlangen.de (R.F.); 2Comprehensive Cancer Center Erlangen-EMN (CCC ER-EMN), 91054 Erlangen, Germany; manuel.schmidt@uk-erlangen.de (M.A.S.); ilker.eyuepoglu@uk-erlangen.de (I.E.); 3Department of Neuroradiology, Universitätsklinikum Erlangen, Friedrich-Alexander-Universität Erlangen-Nürnberg, Schwabachanlage 6, 91054 Erlangen, Germany; 4Department of Neurosurgery, Universitätsklinikum Erlangen, Friedrich-Alexander-Universität Erlangen-Nürnberg, Schwabachanlage 6, 91054 Erlangen, Germany

**Keywords:** meningioma, volumetric analysis, segmentation, transient enlargement, pseudoprogression, stereotactic radiotherapy, radiosurgery, response assessment

## Abstract

**Simple Summary:**

Accurate assessment of treatment efficacy is a prerequisite for the improvement in therapeutic outcomes in clinical trials. However, it is very challenging to accurately track the size of meningiomas after radiotherapy, because of their complex shapes and often slow growth. Measuring the whole tumor volume as opposed to simple diameter measurements to assess treatment efficacy, therefore, is very promising but little is known on expected volumetric changes of meningiomas following radiotherapy. Therefore, in this study, we meticulously investigated volumetric changes in meningiomas following radiotherapy incorporating volumetric measurements from 468 MRI studies and evaluated newly proposed RANO volumetric criteria in the context of radiotherapy. We found that temporary tumor enlargement after radiotherapy overall was rare but occurred significantly more frequently after high than after low single doses of radiation, represented an important differential diagnosis to tumor progression and would have skewed results in a clinical trial if not accounted for.

**Abstract:**

To investigate the occurrence of pseudoprogression/transient enlargement in meningiomas after stereotactic radiotherapy (RT) and to evaluate recently proposed volumetric RANO meningioma criteria for response assessment in the context of RT. Sixty-nine meningiomas (benign: 90%, atypical: 10%) received stereotactic RT from January 2005–May 2018. A total of 468 MRI studies were segmented longitudinally during a median follow-up of 42.3 months. Best response and local control were evaluated according to recently proposed volumetric RANO criteria. Transient enlargement was defined as volumetric increase ≥20% followed by a subsequent regression ≥20%. The mean best volumetric response was −23% change from baseline (range, −86% to +19%). According to RANO, the best volumetric response was SD in 81% (56/69), MR in 13% (9/69) and PR in 6% (4/69). Transient enlargement occurred in only 6% (4/69) post RT but would have represented 60% (3/5) of cases with progressive disease if not accounted for. Transient enlargement was characterized by a mean maximum volumetric increase of +181% (range, +24% to +389 %) with all cases occurring in the first year post-RT (range, 4.1–10.3 months). Transient enlargement was significantly more frequent with SRS or hypofractionation than with conventional fractionation (25% vs. 2%, *p* = 0.015). Five-year volumetric control was 97.8% if transient enlargement was recognized but 92.9% if not accounted for. Transient enlargement/pseudoprogression in the first year following SRS and hypofractionated RT represents an important differential diagnosis, especially because of the high volumetric control achieved with stereotactic RT. Meningioma enlargement during subsequent post-RT follow-up and after conventional fractionation should raise suspicion for tumor progression.

## 1. Introduction

Meningiomas are the most common intracranial tumor [1]. Despite the majority being benign WHO I tumors, meningiomas and their treatment not only cause considerable morbidity but may also be detrimental to patient survival [1,2,3]. This is of course especially true in atypical WHO II and anaplastic WHO III tumors, but also uncontrolled supposedly benign WHO I meningiomas can ultimately prove fatal [1,3,4]. It is therefore an important responsibility for the neuro-oncology community to further systematically improve patient outcome in clinical trials that require precise and standardized imaging-based endpoints. Surgery and stereotactic radiotherapy continue to be the mainstay of treatment in meningiomas, while systemic treatments until now have not shown a clear proof of efficacy [5]. The continued interest in radiotherapy for meningiomas has motivated ongoing prospective trials by the EORTC and NRG/RTOG study groups [6,7,8]. However, meningiomas pose a particular challenge to accurately assessing tumor size, as their often complex geometry and low sphericity make precise and reliable uni- or bidimensional measurements difficult. Moreover, due to their slow rate of growth, particularly small changes in tumor size need to be identified in case of I meningiomas. For this very reason volumetric assessment holds particular promise in meningiomas to accurately track changes in tumor size. The very recently proposed Response Assessment in Neuro-Oncology (RANO) meningioma guideline, therefore, puts a strong emphasis on volumetric criteria for assessment of response and progression in meningioma clinical trials [9]. Although easily underestimated, the accurate assessment of treatment efficacy has to be understood as a sine qua non for any systematic improvement in therapeutic outcomes. While still primarily based on bidimensional measurements, the RANO working group put forth a complete set of volumetric criteria to be evaluated and included in meningioma clinical studies [9]. Originating from the requirements of systemic therapy trials, the RANO guideline identifies volumetric changes after stereotactic radiotherapy as an important area of uncertainty where additional research is needed [9]. In particular, the phenomenon of transient enlargement or pseudoprogression in meningiomas following stereotactic radiation is only poorly characterized but constitutes an important differential diagnosis in case of any volumetric enlargement following radiation [9]. In the present study, we therefore investigate volumetric changes in meningiomas following stereotactic radiotherapy with a particular emphasis on transient enlargement and evaluate the newly proposed volumetric RANO criteria in the context of radiotherapy.

## 2. Materials and Methods

### 2.1. Patient Population

Patients that received stereotactic radiotherapy for meningiomas at our tertiary university hospital between January 2005 to May 2018 were retrospectively identified. From this consecutive cohort, 69 meningiomas in 64 unique patients were selected for further analyses that had macroscopic tumors on baseline MRI, additional post-RT MRI follow-up and that had received no prior radiotherapy. Radiotherapy for each patient in the present cohort had been recommended after a joint interdisciplinary review by experts in neurosurgery, neuroradiology, neuropathology as well as radiation oncology within the framework of an interdisciplinary tumor board.

Clinical and radiotherapy dose parameters were obtained from the electronic patient records and the oncology information system. Each patient was classified according to the co-morbidity, tumor location, patient’s age, tumor size, and symptoms/signs (CLASS) algorithmic scale. The CLASS scale is a validated tool to estimate the risks and benefits of surgery in meningioma patients. Incorporating five major factors (co-morbidity, location, age, size as well as signs and symptoms) and two other factors (prior surgery or radiation, radiographic progression), the CLASS scale assigns each patient a CLASS score group of I, II or III. Patients with CLASS score group of I are recommended for surgery; in CLASS II patients surgery, could be considered but with caution and CLASS III patients should not receive surgery in most circumstances [10].

### 2.2. Radiation Therapy 

All patients were treated with single-session radiosurgery (SRS) or fractionated stereotactic radiotherapy (FSRT) using a linear accelerator-based Novalis^®^ or Novalis-Tx^®^ stereotactic radiotherapy system (BrainLAB, Feldkirchen, Germany). Patient immobilization was achieved using an individually manufactured thermoplastic head mask connected to a stereotactic base frame (BrainLAB, Feldkirchen, Germany). Iplan (BrainLAB, Feldkirchen, Germany) was used for radiotherapy treatment planning and target volume definition [11,12]. In all patients a dedicated planning CT was acquired and rigidly co-registered with the baseline MRI using the Iplan software. The gross target volume (GTV) was segmented in the contrast-enhanced baseline T1-MPRAGE sequence including any adjacent dural hyperintensity (i.e., dural tail). Clinical target volume (CTV) was defined as GTV with an extension of 3–5 mm along the dura and PTV was defined as isotropic CTV expansion by 2 mm. For FSRT, dose was prescribed to the ICRU reference point with the PTV being encompassed by the 95% isodose, for SRS dose was prescribed to the encompassing 80% isodose. Daily stereoscopic X-ray imaging (ExacTrac^®^, BrainLAB, Feldkirchen, Germany) was used for patient positioning with stereoscopic X-ray imaging being repeated after every couch rotation in case of SRS [13]. Eighty-three percent (57/69) of meningiomas were treated with conventionally fractionated stereotactic radiotherapy, while the remaining 17% (12/69) were treated with SRS or hypofractionated stereotactic radiotherapy (Table 1).

### 2.3. Follow-Up and Imaging

Siemens 1.5 or 3 T Tesla MRI scanners at our institution were used for image acquisition. The 3D image datasets used in this study consisted of 160 or 192 contiguous, sagittal or transverse planes of 3-dimensional T1-weighted magnetization-prepared rapid gradient-echo (MPRAGE) images with an isotropic resolution of 1 mm × 1 mm × 1 mm (repetition time [TR] = 1900 ms, echo time [TE] = 3.02 ms, inversion time [TI] = 1100 ms, matrix = 256 × 265, field of view [FoV] = 250, flip angle = 15 degrees or TR = 2200 ms, TE = 2.67 ms, TI = 900 ms, matrix = 256 × 246, FoV = 250, flip angle = 8 degrees). As contrast agents 0.2 mL/kg Dotarem (Guerbet, Sulzbach, Germany) or 0.1 mL/kg Gadovist (Bayer, Leverkusen, Germany) were used.

Patients received MRI at baseline (median of 10 days prior to radiotherapy). A first follow-up MRI was routinely performed 3 months after the end of radiotherapy and subsequently every 6 to 12 months. In case of any enlargement, a follow-up scan was performed after a shortened interval of 3 months. However, due to the retrospective nature of the study, patients received MRI at slightly different points in time after SRT.

### 2.4. Volumetric Analysis

Tumor segmentation was performed using 3D Slicer (version 4.5.0) [14]. The software 3D Slicer is an open-source program that is supported by the National Institutes of Health (NIH) [15] and offers a variety of modules for segmentation, volume statistics and image coregistration. A dedicated segmentation module written in Python was developed and used for this study, that utilizes the built-in modules but accelerates the segmentation process by automating steps that do not require user interaction [16]. In the developed segmentation module, a first semi-automatic segmentation step was performed using the VTK Fast Growcut method [17] as semiautomatic segmentation methods have been shown to decrease inter- and intra-observer variabilities [18,19] and are more time-efficient than manual delineation [20]. After the initial semi-automatic segmentation step, each segmentation was reviewed and corrected manually on a slice-by-slice basis using the built-in editor module in 3D Slicer. Finally, all segmentations were additionally reviewed, corrected, and validated by a second expert. Response assessment was performed according to recently proposed volumetric RANO criteria for meningioma clinical trials with progressive disease (PD) being defined as ≥40% increase in volume relative to nadir or baseline (i.e., relative volume ≥140%), minor response (MR) being defined as ≥40% and partial response (PR) being defined as ≥65% decrease in volume relative to baseline (i.e., relative volume ≤60% and ≤35%, respectively). All other lesions were classified as stable (SD) [9]. Change in distant lesions, corticosteroid use or clinical status were not considered in the definition of local progression in the present study. Transient enlargement was defined as ≥20% increase in volume followed by a spontaneous volumetric regression of at least 20% relative to maximum tumor volume, as this is commonly considered to be the lowest volumetric difference that can be reliably detected [21,22].

### 2.5. Statistical Analysis

Local control was calculated from the start of stereotactic radiotherapy to the date of volumetric progression or censored at last follow-up and estimated using the Kaplan–Meier method. Intergroup differences in continuous variables were assessed using the Wilcoxon rank-sum test and differences in categorical parameters were evaluated using Fisher’s exact test. All tests were conducted two-sided. Univariate and multivariate logistic regression analysis was used to evaluate potential predictors for transient enlargement/pseudoprogression. Parameters with univariate *p* < 0.300 were selected for the multivariate model. All statistical analyses were performed using IBM SPSS 21 (IBM, New York, NY, USA) Graphs were created with GraphPad Prism 8.4 (GraphPad, San Diego, CA, USA) and SPSS. *p*-values < 0.05 were considered statistically significant.

## 3. Results

Seventy-five percent (52/69) of meningiomas had imaging diagnosis of low-grade meningioma, while 15% (10/69) and 10% (7/69) had histologic diagnosis of WHO grade I and II meningioma, respectively. Eighty-three percent (57/69) were treated for primary diagnosis while 17% (12/69) were treated for macroscopic recurrence following primary resection. As part of treatment, 7% (5/69) had preceding Simpson Grade IV/V resection in addition to radiotherapy. Median age of the cohort was 63 years (range, 49.5–72) and sex was female in 75% (52/69). Most common locations were parasellar (30%), convexity (26%) and tentorial (20%). Most cases had a CLASS score group of II or III (96%) reflecting an unfavorable risk–benefit ratio for surgical treatment in the studied cohort (Table 1) [10].

Median imaging follow-up following stereotactic radiotherapy (SRT) was 42.3 months and an average of 6.8 MRI studies (IQR, 4–9) were available per case for volumetric analysis. Mean best volumetric response was −23% volume change from baseline (range, −86% to +19%, IQR, −33% to −7%). According to volumetric RANO criteria, best response was stable disease (SD) in 81% (56/69), minimal response (MR) in 13% (9/69) and partial response (PR) in 6% (4/69). Interestingly, meningiomas treated with SRS or hypofractionated radiotherapy showed significantly more pronounced volumetric regression than those treated with conventionally fractionated radiotherapy (mean best change from baseline, −41% vs. −19%, Wilcoxon rank-sum *p* = 0.003). Best volumetric response was observed in 46% (32/69) at last imaging follow-up reflecting that most lesions were characterized by a slow but ongoing volumetric regression after radiotherapy (Figure 1 and Figure 2).

Examining the relationship between WHO grade (low-grade vs. WHO °II) and the RANO volumetric best response category, no statistical significance could be found (Fisher’s exact test *p* = 0.514). In addition, when looking at best volumetric regression during follow-up in a continuous fashion, no difference between low-grade and WHO °II meningiomas could be found (mean best response, −22.7% vs. −23.2%, Wilcoxon rank-sum *p* = 0.905).

Transient enlargement after radiotherapy occurred in only 6% (4/69) of meningiomas (Figure 1 and Figure 2). Transient enlargement was characterized by a mean maximum volume increase of +181% from baseline (range, +24% to +389 %) with all cases of transient enlargement occurring in the first year after radiation (mean of 7.8 months post-RT, range 4.1 to 10.3 months). After increasing by ≥40% of baseline volume, all cases of transient enlargement had spontaneously regressed by ≥20% on next imaging follow-up, which was conducted after a median of 3.0 months (range, 3.0–5.8 months). Interestingly, transient enlargement was significantly more frequent in meningiomas treated with SRS or hypofractionated RT than in meningiomas that received conventionally fractionated RT (25% vs. 2%, Fisher’s exact test *p* = 0.015). When looking at the impact of WHO grade (low-grade vs. WHO °II) on transient enlargement, the rate of transient enlargement was not significantly different between low-grade and WHO °II meningiomas (Fisher’s exact test *p* = 0.355). In multivariate logistic regression analysis, only the type of radiotherapy (hypofractionation or SRS vs. conventional fractionation) was a significant predictor for pseudoprogression/transient enlargement (odds ratio 22.53, *p* = 0.041) (Table 2).

Sixty percent (3/5) of meningiomas that reached the volumetric threshold for progressive disease according to RANO, had only transient enlargement/pseudoprogression followed by a spontaneous and often marked volumetric decline (Figure 1 and Figure 2). Therefore, while transient enlargement/pseudoprogression overall was rare, it constituted an important differential diagnosis in cases of volumetric enlargement. Consequently, for the overall cohort, 5-year local control was 92.9% when pseudoprogression was not recognized, while it reached 97.8% if taken into account (Figure 3). Differentiating by grade, 5-year apparent local control would have been 96.3% compared to 100.0% when looking exclusively at low-grade meningiomas. In grade °II meningiomas, 5-year local control would have been 68.6%, if pseudoprogression was not recognized compared to 80.0% if pseudoprogression was accounted for.

## 4. Discussion

Transient enlargement or pseudoprogression is an important consideration during imaging follow-up in many neuro-oncologic entities treated with stereotactic radiation. This transient increase in tumor size, frequently indiscernibly mimicking true tumor progression, is especially prevalent and well-described in malignant gliomas [23,24,25]. In these malignant tumors, pseudoprogression commonly occurs in the first 12 weeks following chemoradiation with temozolomide, especially when MGMT promotor methylation is present and generally indicates a more favorable prognosis [23,24]. In glioblastoma, pseudoprogression has been observed in as much as 31% of patients one month after concurrent chemoradiation in a study by Brandes et al. [23]. However, benign tumors have also been shown to develop transient enlargement following radiotherapy. In vestibular schwannomas, Meijer et al. observed transient swelling in 30% of cases following SRS, whereas Aoyama et al. reported transient enlargement in 14% of tumors following conventionally fractionated radiotherapy [26,27]. Knowledge on expected volumetric changes and the incidence of transient enlargement following radiotherapy are crucial for clinical practice to guide decision making in case of an enlarging lesion but are also needed to define suitable criteria for assessment in clinical trials. However, despite their high prevalence, only few studies have investigated the phenomenon of transient enlargement in meningiomas following radiotherapy before. Kim et al. reported transient enlargement in only 0.8% of meningiomas after SRS as an incidental finding and Miyatake et al. found a transient volumetric increase in 23% of grade III meningiomas following boron neutron capture therapy [28,29]. Recently, Fega et al. described transient enlargement in two out of nine meningiomas (22%) treated with hypofractionated radiotherapy [30]. Because of this lack of evidence, the recently proposed RANO meningioma guideline identified transient enlargement following radiotherapy as an area that is poorly characterized and requires further investigation [9]. In the present volumetric study, we found a low overall incidence of transient enlargement in meningiomas of 6%, that is considerably smaller than the frequencies commonly described for vestibular schwannomas post RT [26,27]. However, because of the high rate of local control during the available follow-up period, transient enlargement constituted an important differential diagnosis for enlarging lesions (60% of meningiomas that reached the threshold for progression) and could have considerably skewed results in case of a clinical trial, if not accounted for. A particularly important finding was that transient enlargement in the present study was significantly more common following SRS or hypofractionation than after conventional fractionation (25% vs. 2%, *p* = 0.015). The high incidence of 25% transient enlargement in the SRS/hypofractionated group is very similar to the values reported by Fega et al. for hypofractionated RT with 5 × 5 Gy (22%) and for boron neutron capture therapy by Miyatake et al. (23%) [28,30]. Conversely, because of the low frequency of transient enlargement following conventional fractionation, an increase in meningioma size after treatment with low single doses is much more likely to represent true tumor progression. Similarly, since all cases of transient enlargement in the present study occurred within one year after radiotherapy, enlargement during later imaging follow-up should also raise suspicion for progression. After reaching the volumetric RANO threshold for progression, all cases of transient enlargement had spontaneously regressed by ≥20% on next imaging follow-up during a median interval of 3.0 months. This finding supports the practice of obtaining a follow-up MRI scan after 3 months when initially observing post-RT enlargement to differentiate transient enlargement from true progression, that is also proposed by the current RANO recommendation [9].

An additional aim of the present study was to evaluate the newly proposed volumetric RANO criteria in the context of radiotherapy, which had been primarily developed for the purpose of systemic therapy trials [9]. The volumetric RANO meningioma criteria overall were well-suited for the assessment of response and progression following stereotactic radiation if transient enlargement was accounted for. According to the volumetric RANO criteria, best response was minimal response in 13% and partial response in 6% in the present study, reflecting that radiotherapy mostly achieved sustained tumor control and, more rarely, profound regression in this cohort dominated by low-grade meningiomas. The fact that best volumetric response was observed at the date of last imaging in 46% of cases, however, indicates that additional volumetric regression probably would have been observed with additional imaging follow-up. The observed mean best volumetric response of −23% volume change from baseline in the present study after a median follow-up of 42.3 months was consistent with previous studies conducted before the publication of standardized volumetric guidelines [31,32,33,34,35,36]. An interesting and unprecedented finding was that meningiomas regressed significantly more with SRS and hypofractionation than with conventional fractionation, which might reflect differential biologic effects of high and low single doses of radiation like an increased targeting of endothelial cells with hypofractionation and SRS [37,38]. An additionally interesting research question is the potential differences in volumetric response in higher-grade compared to low-grade meningiomas. As faster-growing tumors usually show an increased response to radiotherapy, whether higher-grade meningiomas show more profound volumetric regression is a very plausible hypothesis to test. In the present study, we were not able to identify differences in regard to best volumetric response or transient enlargement between low-grade and WHO °II meningiomas. However, as the proportion of WHO II meningiomas was low in the present cohort, this analysis was considerably underpowered. A potential difference in volumetric response between low- and higher-grade meningiomas, therefore, is an interesting research question that should be addressed in future retrospective and prospective multicenter studies.

The distribution of fractionation schemes with a predominance of conventional fractionation in our study reflects that meningiomas, because of their often large size, are frequently treated with conventional single doses. In fact, current treatment protocols by the RTOG and EORTC both use conventional fractionation in meningiomas [6,7]. It is therefore particularly important to investigate volumetric changes for this most commonly used fractionation scheme. Conversely, the total number of cases treated with SRS and hypofractionation was low in the present study (*n* = 12), which constituted a limitation.

Despite follow-up intervals of 10 years and even 25 years would be needed to definitely evaluate long-term control in low-grade meningiomas [2,4], the achieved 5-year local volumetric control of 97.8% in the present study that is further supported by the individual longitudinal volumetric data was excellent (Figure 1 and Figure 2, Appendix A). Considering that all cases had a macroscopic tumor at baseline and only a minority (7%) had undergone partial resection prior to radiotherapy, the present study provides additional evidence that radiotherapy alone achieves a favorable outcome in meningioma patients suitable and unsuitable for preceding surgery. 

### Limitations

The number of cases was a limitation in the present study, especially in the SRS and hypofractionated subgroup. Due to the retrospective nature, the timing of imaging was not strictly standardized and the cohort was less homogeneous. A main limitation was the lack of clinical information with toxicity of treatment not having been assessed. As this study was devoted to investigating volumetric changes in contrast-enhanced 3D T1w MRI sequences, perifocal edema and changes in T2w sequences were not evaluated. However, taking these limitations into account, the study adds important findings to the literature that are important for daily clinical practice and standardized volumetric assessment in clinical trials. 

## 5. Conclusions

Transient enlargement in meningiomas in the first year following SRS and hypofractionated RT represented an important differential diagnosis to true tumor progression in the present study. Transient enlargement was rare after conventional fractionation and no case of transient enlargement was observed beyond one year of follow-up; thus, meningioma enlargement in these situations should raise suspicion for true tumor progression. Transient enlargement following radiotherapy needs to be addressed in meningioma clinical trials, especially if radiation is compared to systemic treatments or observation. The newly proposed volumetric RANO meningioma criteria were well-suited to assess local control, and 5-year volumetric control was excellent following stereotactic radiotherapy if transient enlargement was accounted for.

## Figures and Tables

**Figure 1 cancers-14-01547-f001:**
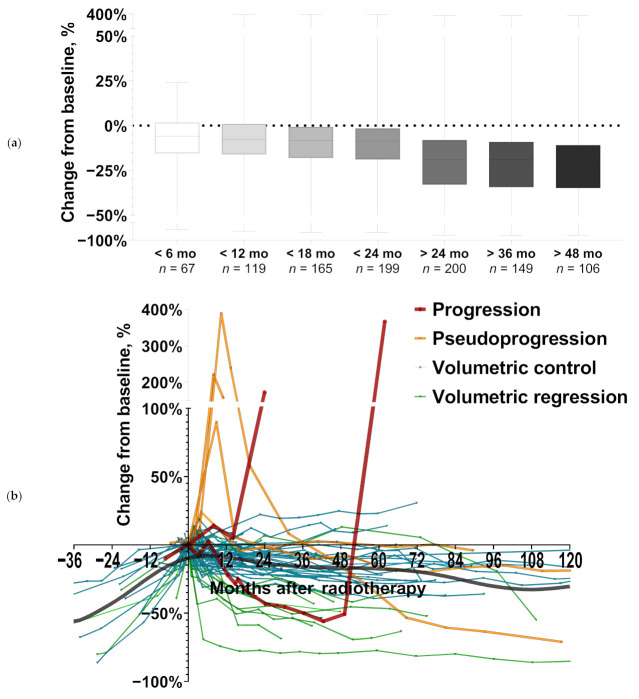
(**a**) Boxplot illustrating mean volumetric decrease over time following radiotherapy. Whiskers: minimum to maximum. (**b**) Spider plot illustrating tumor volume changes over time following radiotherapy (RT) for all cases. Dark red: cases with progression during post-RT follow-up; orange: transient enlargement/pseudoprogression; cyan: volumetric control; green: volumetric regression (i.e., minor and partial response according to volumetric RANO criteria). Local regression curve (dark gray line, LOWESS) indicates overall trend in volumetric changes following stereotactic radiotherapy.

**Figure 2 cancers-14-01547-f002:**
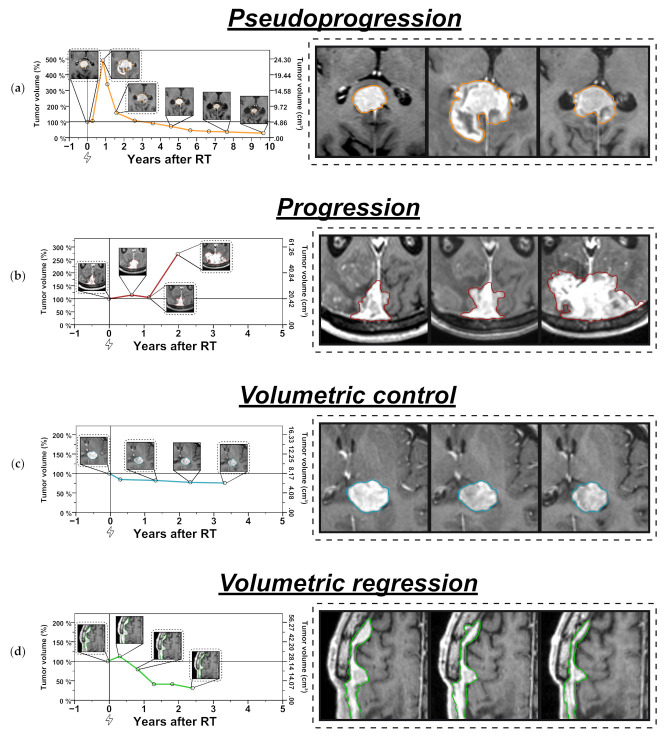
(**a**) Individual examples of pseudoprogression/transient enlargement, (**b**) progression, (**c**) volumetric control and (**d**) volumetric regression. Graphs on the left side: meningioma volumes are expressed relative to baseline volume (left y-axis) over time with the right y-axis showing the absolute tumor volume in cm^3^. The flash symbol indicates the time of radiotherapy (0 months—100% relative tumor volume). Inlay images show tumor segmentations for different measurement time points. Images on the right side: enlarged view of selected segmentations highlighted in the corresponding left-hand graph. Scale bar or magnification.

**Figure 3 cancers-14-01547-f003:**
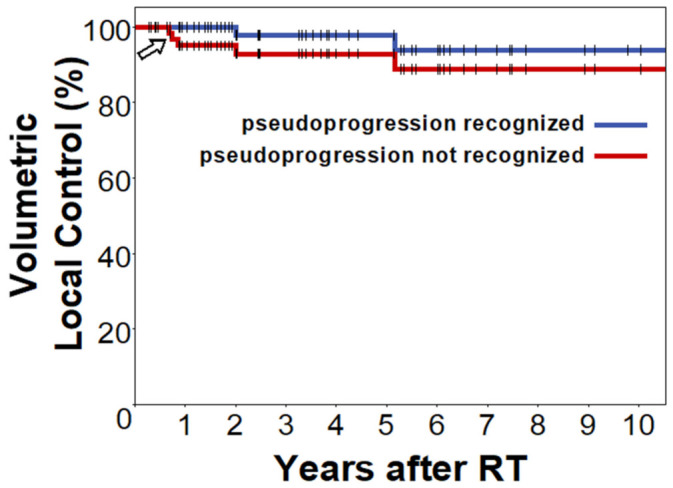
Apparent volumetric local control, if the study would have been a prospective trial and pseudoprogression was accounted for (blue) and was not accounted for (red). Kaplan–Meier plots with vertical bars indicating censored cases. In the red plot, surpassing the volumetric threshold for progression was counted as an event (step in Kaplan–Meier plot) irrespective of further follow-up. In the blue plot, surpassing the threshold for volumetric progression was not counted as an event, if followed by spontaneous volumetric regression ≥20% relative to maximum tumor volume (=pseudoprogression). In the case of a prospective trial, this corresponds to continuing imaging follow-up beyond surpassing the threshold for volumetric progression and incorporating pseudoprogression/transient enlargement in the response assessment criteria. Note: additional progression events in the first year after RT, if transient enlargement/pseudoprogression was not recognized (steps in Kaplan–Meier plot, arrow).

**Table 1 cancers-14-01547-t001:** Characteristics of treated meningiomas (N = 69).

Meningioma Characteristic	Total Cohort (N = 69)
Patient age, years	
Median (IQR)	63.0 (49.5–72.0)
Mean (range)	61.0 (36–86)
Sex, *n* (%)	
Male	17 (25%)
Female	52 (75%)
Primary indication for radiotherapy, *n* (%)	
Tumor progression/recurrence on imaging	33 (48%)
Tumor-related signs/symptoms	28 (41%)
Residual tumor after preceding surgery	5 (7%)
Patient request	3 (4%)
Primary diagnosis vs. recurrence, *n* (%)	
Primary diagnosis	57 (83%)
Recurrence after prior resection	12 (17%)
Preceding surgery, *n* (%)	
Preceding Simpson grade IV/V resection	5 (7%)
No preceding resection	64 (93%)
Histology, *n* (%)	
WHO I	10 (15%)
WHO II	7 (10%)
Imaging diagnosis of low-grade meningioma	52 (75%)
Meningioma CLASS score group, *n* (%)	
Group I	3 (4%)
Group II	23 (33%)
Group III	43 (62%)
Location, *n* (%)	
Parasellar	21 (30%)
Convexity	18 (26%)
Tentorial	14 (20%)
Olfactory	5 (7%)
Sphenoid Wing	5 (7%)
Falcine	2 (3 %)
Parasagittal	2 (3 %)
Foramen magnum	1 (1%)
Ventricular	1 (1%)
Pre-RT baseline volume, cm^3^	
Median (IQR)	4.7 (1.3–9.0)
Mean (range)	7.7 (0.2–43.1)
RT fractionation scheme, *n* (%)	
Conventional fractionation in single doses of 1.8 Gy	57 (83%)
10 × 4.0 Gy	1 (1%)
7 × 5.0 Gy	1 (1%)
1 × 13.0 Gy	5 (7%)
1 × 14.0 Gy	5 (7%)
Conventional fractionation total dose, Gy	
Median (IQR)	54.0 (54.0–54.0)
Mean (range)	53.8 (50.4–59.4)
Number of post-RT imaging studies excluding baseline	
Median (IQR)	5 (3.0–8.0)
Mean (range)	5.8 (1.0–15.0)

Sixty-nine meningiomas in 64 unique patients were treated with stereotactic radiotherapy and followed volumetrically.

**Table 2 cancers-14-01547-t002:** Logistic regression analysis of predictive factors for pseudoprogression/transient enlargement (*n* = 69).

Parameter	Univariate		Multivariate	
	OR	*p*-Value	OR	*p*-Value
SRS/hypofractionation vs. conventional fractionation	18.67	0.016	22.53	0.041
Baseline tumor volume, ≥4.7 vs. <4.7 cm^3^	0.29	0.289	1.40	0.831
WHO grade (atypical vs. benign)	3.28	0.335	Not included because of *p* ≥ 0.300	
Patient age, ≥63 vs. <63 years	0.91	0.929	Not included because of *p* ≥ 0.300	
Recurrence vs. primary diagnosis	1.64	0.682	Not included because of *p* ≥ 0.300	
Patient sex (male vs. female)	1.02	0.986	Not included because of *p* ≥ 0.300	
Preceding surgery	0.00	0.999	Not included because of *p* ≥ 0.300	

Univariate and multivariate logistic regression analysis of predictive factors for pseudoprogression/transient enlargement. Parameters with univariate *p* < 0.300 were included in the multivariate model. OR: Odds ratio.

## Data Availability

Data are available upon request. However, legal restrictions, especially the EU General Data Protection Regulation (GDPR), the German Data Protection Laws and the Bavarian Hospital law apply, so some requests may have to be declined partially or completely.

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
