# Peer review of "Transient Enlargement in Meningiomas Treated with Stereotactic Radiotherapy"

_cancers, 2022, doi:10.3390/cancers14061547_

Round 1

Reviewer 1 Report

I congratulate the authors for this well written and very interesting manuscript. The main message is that despite the existing RANO criteria, meningiomas should be well monitored after radiotherapy and in some cases true progression must be distinguished from pseudoprogression with longer observation, as there are some cases with transient tumor enlargement.

Nevertheless, I have some comments regarding minor revisions.

1) In table 1 you should give insight of the indications for radiotherapy (e.g. tumor progression, symptoms, patients wish , …)

2) I consider Figure 1 a) to be very confusing. As a rough overview, it can be used in the publication. In my opinion, however, the mean volume decrease should also be shown in a figure. In this regard, I have uploaded an example. I recommend box whisker plots here. They can show the volume change best. It would then be necessary to add how many patients were still included in the study at the timepoints of the measurements. For instance, I assume that at the timepoint 24 months after irradiation there were significantly more follow-up MRI examined than at timepoint 48 months after irradiation.

3) You found out that 60% of patients had transient volume enlargement. Was this influenced by the WHO grade? Was volume regression after radiotherapy in WHO II more often? I would like to have a comment on that in the results section.

4) Line 257-260: “According to the volumetric RANO criteria, best response was Stable Disease in 81%, Minimal Response in 13% and Partial Response in 6% in the present study, reflecting that radiotherapy mostly achieves sustained tumor control and more rarely profound regression in lower-grade meningiomas.”

Here, you repeat your results which is redundant and additionally you also have 10% higher-grade meningiomas in your collective. I think you should differentiate here.

5) Line 276-277: “… the present study provides additional evidence that radiotherapy alone achieves favorable outcome in meningioma patients unsuitable for preceding surgery.”

In your table 1 26% had meningiomas in the convexity. Meningiomas in this region are normally easy to resect. You have to change this sentence: “… the present study provides additional evidence that radiotherapy alone achieves favorable outcome in meningioma patients suitable and unsuitable for preceding surgery.”

6) In section 4.1. you should mention that a main limitation of your study is the lack of clinical information. Furthermore, toxicity of the treatment (e.g. perifocal edema) was not assessed.

Reviewer 2 Report

P05L172: we find difficult to read the number of month on figure 1. (a)

Could you please make it clearer or bigger?

Reviewer 3 Report

I am reviewing the article “Transient enlargement in meningiomas treated with stereotactic radiotherapy”. (cancers-1586877). This study is to investigate the occurrence of pseudoprogression/transient enlargement in meningiomas after stereotactic radiotherapy (RT) and to evaluate recently proposed volumetric RANO meningioma criteria for response assessment in the context of RT. A total of 468 MRI studies were segmented longitudinally during a median follow-up of 42.3 months. Transient enlargement was characterized by a mean maximum volumetric increase of +181% (range, +24% to +389 %) with all cases occurring in the first year post-RT (range, 4.1-10.3 months). The topic is actually not new, but detail to discuss the volume response following RT.  The article can be accepted due to the set of study design is comprehensive and meaningful. 

One question, 
Page 4: Response assessment was performed according to recently proposed volumetric RANO criteria for meningioma clinical trials with progressive disease (PD) being defined as ≥40% increase  in volume relative to nadir or baseline, minor response (MR) being defined as ≥40% and partial response (PR) being defined as ≥65% regression relative to baseline

Please make sure the number and the direction of “<“ and ”>” were right. 

Reviewer 4 Report

Title:

The authors entitled the article “Transient enlargement in meningiomas treated with stereotactic radiotherapy”, but vast majority of patients were treated with 1.8 Gy per fraction. This is misleading and does not reflect the body of the article.

Materials and Methods:

  • Patients were retrospectively identified that received stereotactic radiotherapy for 85 meningiomas at our tertiary university hospital between 01/2005 to 05/2018: according to table 1, 83% of patients have been treated with conventional fractionation (1.8 Gy per fraction). I think that the term “stereotactic” has been used in a misleading way in title, abstract and materials and methods sections.

Results:

  • Median imaging follow-up following stereotactic radiotherapy (SRT) was 42.3 162 months and an average of 6.8 MRI studies (IQR, 4 – 9):in methods section, authors underlined that imaging was performed every 3 months, but patients received MRI at slightly different points in time after SRT. Here, 3 months is outside from the reported IQR, indicating that evaluation was performed outside the three months period for the vast majority of patients. I think that materials and methods section should be revised accordingly
  • Interestingly, meningiomas treated with SRS or 167 hypofractionated radiotherapy showed significantly more pronounced volumetric regres- 168 sion than those treated with conventionally fractionated radiotherapy (Mean best change 169 from baseline, -41% vs. -19%, Wilcoxon rank-sum p = 0.003).: Again, the title and the methods section underlined that all patients were treated with stereotactic radiotherapy, but then a subgroup analysis about conventionally fractionated radiotherapy and hypofractionated radiotherapy is reported.
  • 60% (3/5) of meningiomas that reached the volumetric threshold for progressive disease according to RANO, had only transient enlargement followed by a spontaneous and often marked volumetric decline: One of the main conclusion of the manuscript (“Transient enlargement/pseudoprogression in the first year following SRS and hypofractionated RT represents 43 an important differential diagnosis”) is based on an overall population of 5 patients progressing after RT. I think that numbers are limited to draw any conclusion.
